IPPP/20/97

# Elastic photon–initiated production at the LHC: the role of hadron–hadron interactions

L.A. Harland–Lang[1], V.A. Khoze[2,3], M.G. Ryskin[3]

[1]Rudolf Peierls Centre, Beecroft Building, Parks Road, Oxford, OX1 3PU
[2]Institute for Particle Physics Phenomenology, University of Durham, Durham, DH1 3LE
[3]Petersburg Nuclear Physics Institute, NRC Kurchatov Institute, Gatchina, St. Petersburg, 188300, Russia

## Abstract

We analyse in detail the role of additional hadron–hadron interactions in elastic photon–initiated (PI) production at the LHC, both in $pp$ and heavy ion collisions. We first demonstrate that the source of difference between our predictions and other results in the literature for PI muon pair production is dominantly due to an unphysical cut that is imposed in these latter results on the dimuon–hadron impact parameter. We in addition show that this is experimentally disfavoured by the shape of the muon kinematic distributions measured by ATLAS in ultraperipheral PbPb collisions. We then consider the theoretical uncertainty due to the survival probability for no additional hadron–hadron interactions, and in particular the role this may play in the tendency for the predicted cross sections to lie somewhat above ATLAS data on PI muon pair production, in both $pp$ and PbPb collisions. This difference is relatively mild, at the $\sim 10\%$ level, and hence a very good control over the theory is clearly required. We show that this uncertainty is very small, and it is only by taking very extreme and rather unphysical variations in the modelling of the survival factor that this tension can be removed. This underlines the basic, rather model independent, point that a significant fraction of elastic PI scattering occurs for hadron–hadron impact parameters that are simply outside the range of QCD interactions, and hence this sets a lower bound on the survival factor in any physically reasonable approach. Finally, other possible origins for this discrepancy are discussed.

## 1 Introduction

The LHC is a collider of protons and heavy ions, both of which are electromagnetically charged objects; hence, as well as being a QCD machine, it can act as a source of photons. Such photon–initiated (PI) processes are a key ingredient in the LHC precision physics programme, providing a unique probe of physics within and beyond the SM, see e.g. [1] for further discussion and references, and [2–6] for reviews.

In such PI interactions the colour singlet nature of the photon naturally leads to exclusive events with intact hadrons in the final state. In the case of proton–proton ($pp$) collisions, this opens up the exciting possibility of measuring the outgoing intact protons using dedicated forward proton detectors at the LHC, namely the AFP [7, 8] and CT–PPS [9] detectors, which have been installed in association with both ATLAS and CMS, respectively. These detectors have most recently been used in a measurement of lepton pair production with a single proton tag by ATLAS [10] (the first evidence for which was presented by CMS–TOTEM in [11]) and to place limits on anomalous gauge couplings in the diphoton final state with both protons tagged by CMS–TOTEM [12]. As described in detail in [13], an exciting and broad range of

measurements is also possible during HL–LHC running. Even without tagged protons, one can still select events due to PI production by requiring that rapidity gaps are present in the final state. Indeed a range of data on PI lepton and $W$ boson pair production has been taken at the LHC using this method, by both ATLAS [14–16] and CMS [17–19]. In such a case, both elastic and inelastic photon emission will in general contribute, see [1] for recent theoretical discussion of this.

The possibilities for PI production are not limited to proton–proton collisions, however. In heavy ion collisions, the flux of photons emitted by the colliding hadrons is enhanced by $\sim Z^2$ for each beam in comparison to the proton case and hence the rate for PI production of lower mass objects can be enhanced. In PbPb collisions, data on light–by–light scattering, as well as corresponding constraints on axion–like particles (ALPs), have been presented by both ATLAS [20, 21] and CMS [22], while measurements of dilepton production in the continuum region have been presented at the LHC by ALICE [23] and ATLAS [24].

A key element in the above processes is that the initiating photons must have rather low virtuality, $Q^2$, in order for the photon to be emitted elastically from the hadron. Considering the interaction in terms of the impact parameter of the colliding hadrons, this corresponds to rather large transverse separations, where the probability of QCD interactions between the hadrons is low. This is discussed in e.g. [1, 25] for the case of $pp$ collisions, while in PbPb collisions it is well established, and indeed we talk about 'ultraperipheral' heavy ion collisions precisely for this reason. The upshot is that to first approximation one can talk about using the LHC as a photon–photon collider; we expect the 'survival factor', $S^2$, i.e. the probability for no additional inelastic hadron–hadron interactions, to be relatively close to unity, and hence the sensitivity to QCD effects to be low.

However, the above consideration is indeed only a first approximation. In reality the survival factor is close to unity, but is not exactly so, and there is some non–negligible probability for hadron–hadron interactions that we must account for, in particular if e.g. precision BSM constraints are being aimed for. This is discussed in [1] for the case of $pp$ collisions, where a precise differential account of the survival factor is presented. In this work, we found that indeed $S^2 \sim 70 - 100\%$ for purely elastic and single dissociative PI lepton pair production, while for double dissociative production (i.e. with no intact protons) it is much lower, with $S^2 \sim 10\%$; however, the precise value depends on the process and the particular event kinematics. In PbPb collisions, we have presented an analysis in [26], and the survival factor for e.g. PbPb collisions is found to be $\sim 70 - 80\%$, again with the precise value depending on the process and kinematics. For both processes, PI production is implemented in the publicly available `SuperChic` MC generator [27].

Interestingly, there is evidence from ATLAS data on muon pair production in both $pp$ [14,16] and, as we will show, PbPb [24] collisions, that the baseline `SuperChic` predictions overshoot the measured cross sections by $\sim 10\%$ (corresponding to a $\sim 2 - 4\sigma$ excess). The most recent data with a single proton tag [10] is consistent within $2\sigma$, but also lies below the `SuperChic` prediction, albeit within rather large experimental errors. Moreover, the $pp$ data [14,16], corrected back experimentally to elastic production cross sections, are apparently better described by the predictions of [28], while the predictions of the `STARlight` MC generator [29] are in fact found to undershoot the PbPb data, i.e. to be rather lower than the `SuperChic` results. Given these findings, two natural questions to ask are: first, what is the reason for these differences, both between theoretical implementations and in the data/theory comparison; second, given this, what are the theoretical uncertainties on these predictions, and is the data/theory comparison improved when these are accounted for?

In this paper, we will address both questions. We will in particular show that the dominant reason for the differences between our results and those of [28, 29] is due to an unphysical cut

on the dimuon–hadron impact parameter, $b_{i\perp}$, that is applied in [28, 29], which require that $b_{i\perp} > R_A$, where $R_A$ is the hadron radius. This effectively assumes that the produced muons and the hadrons will interact inelastically, leading to hadron break up and colour flow between the colliding particles, if their impact parameter lies in this region. This may be reasonable for the production of hadrons, but will not be here. In principle additional QED exchanges between the lepton pair and the ions can play a role, but the impact of this should not be accounted for according to such a procedure. In particular, these higher order QED effects will not be localised in such a way, and will not lead to colour flow between the hadrons at all, and certainly not with unit probability in this region, as such a cut implies. This point has been discussed from a theoretical point of view in [1, 30–32], but interestingly in the ATLAS PbPb data [24] there is clear evidence that the shape of the `STARlight` MC predicted distributions with respect to the muon kinematic variables do not match the data. Indeed, it is suggested in [24] that a loosening of the above requirement may improve the agreement. We will show that imposing this unphysical requirement in the `SuperCHIC` implementation induces a change in the predicted distributions that closely matches the discrepancy between `STARlight` and the data, and hence that without imposing this requirement we can expect a significantly improved description. In other words, as well as being disfavoured theoretically, we demonstrate here that it is disfavoured experimentally.

Once this requirement is removed, however, the predicted cross section is automatically larger. Indeed, we will show that when this restriction is imposed in the `SuperChic` MC predictions, these become rather similar to those of [28, 29]. In other words, this is indeed the principle cause of the difference between these results, and once it is removed these will overshoot the ATLAS dimuon data in both the *pp* and PbPb cases. Given this, we also consider the second question described above in detail, namely what are the theoretical uncertainties in these predictions, and is the data/theory comparison consistent within these? We will in particular consider in detail the naively most obvious source of theoretical uncertainty, due to the modelling of the survival factor. We find that reasonable model variations within the approach of `SuperChic` (based on the formalism described in e.g. [33]) only affect the predictions at the $\lesssim 1\%$ level, and similarly for uncertainties in the underlying hadron EM form factors. Hence we expect the theoretical uncertainty due to the survival factor to be small, and this cannot account for the apparent discrepancy between data and theory.

One may nonetheless question the model dependence of such a statement. To clarify this further we in addition consider very extreme variations in the evaluation of the survival factor. We will show in particular that it is only by including a survival probability that corresponds to the case of inelastic hadron–hadron interactions occurring with unit probability out to impact parameters $b_{i\perp} \sim 3R_A$ that the ATLAS data begins to be matched by the predictions. For PbPb collisions in particular, this separation is beyond the reach of QCD. This underlines the basic, rather model independent, point that a significant fraction of elastic PI scattering occurs for hadron–hadron impact parameters that are simply outside the range of QCD interactions, and hence this sets a lower bound on the survival factor in any physically reasonable approach. Given this, we will also briefly review other potential sources of uncertainty, due to higher order QED effects in PbPb case, and final–state photon emission in both the *pp* and PbPb cases.

The outline of this paper is as follows. In Section 2.1 we present a brief recap of the theoretical framework used to calculate PI production at the LHC. In Section 2.2 we discuss how the $b_{i\perp} > R_A$ cut can be implemented within our calculation. In Section 3 we present results for the impact of this on ATLAS *pp* and PbPb data. In Section 4 we discuss the theoretical uncertainties on these predictions, focussing on the survival factor. Finally, in Section 5 we conclude.

## 2 Theory

### 2.1 Elastic photon–initiated production in hadron collisions: recap

The basic formalism follows that described in for example [26]. That is, the elastic photon–initiated cross section in $N_1 N_2$ collisions is given in terms of the equivalent photon approximation (EPA) [34] by

$$\sigma_{N_1 N_2 \to N_1 X N_2} = \int \mathrm{d}x_1 \mathrm{d}x_2\, n(x_1) n(x_2) \hat{\sigma}_{\gamma\gamma \to X} \ , \tag{1}$$

where $N_i$ denotes the parent particle, and the photon flux is

$$n(x_i) = \frac{\alpha}{\pi^2 x_i} \int \frac{\mathrm{d}^2 q_{i\perp}}{q_{i\perp}^2 + x_i^2 m_{N_i}^2} \left( \frac{q_{i\perp}^2}{q_{i\perp}^2 + x_i^2 m_{N_i}^2} (1 - x_i) F_E(Q_i^2) + \frac{x_i^2}{2} F_M(Q_i^2) \right) \ , \tag{2}$$

in terms of the transverse momentum $q_{i\perp}$ and longitudinal momentum fraction $x_i$ of the parent particle carried by the photon. The modulus of the photon virtuality, $Q_i^2$, is given by

$$Q_i^2 = \frac{q_{i\perp}^2 + x_i^2 m_{N_i}^2}{1 - x_i} \ . \tag{3}$$

For the proton, we have $m_{N_i} = m_p$ and the form factors are given by

$$F_M(Q_i^2) = G_M^2(Q_i^2) \qquad F_E(Q_i^2) = \frac{4 m_p^2 G_E^2(Q_i^2) + Q_i^2 G_M^2(Q_i^2)}{4 m_p^2 + Q_i^2} \ , \tag{4}$$

with

$$G_E^2(Q_i^2) = \frac{G_M^2(Q_i^2)}{7.78} = \frac{1}{\left(1 + Q_i^2/0.71 \mathrm{GeV}^2\right)^4} \ , \tag{5}$$

in the dipole approximation, where $G_E$ and $G_M$ are the 'Sachs' form factors. In this work we do not use the dipole approximation but rather, as in [1], the fit from the A1 collaboration [35].

For the heavy ion case the magnetic form factor is only enhanced by $Z$, and so can be safely dropped. We then have

$$F_M(Q_i^2) = 0 \qquad F_E(Q_i^2) = F_p^2(Q_i^2) G_E^2(Q_i^2) \ , \tag{6}$$

where $F_p^2(Q^2)$ is the squared form factor of the ion. Here, we have factored off the $G_E^2$ term, due to the form factor of the protons within the ion; numerically this has a negligible impact, as the ion form factor falls much more steeply, however we include this for completeness. The ion form factor is given in terms of the proton density in the ion, $\rho_p(r)$, which is well described by the Woods–Saxon distribution [36]

$$\rho_p(r) = \frac{\rho_0}{1 + \exp\left[(r - R)/d\right]} \ , \tag{7}$$

where the skin thickness $d \sim 0.5 - 0.6$ fm, depending on the ion, and the radius $R \sim A^{1/3}$. The density $\rho_0$ is set by requiring that

$$\int \mathrm{d}^3 r\, \rho_p(r) = Z \ . \tag{8}$$

The ion form factor is then simply given by the Fourier transform

$$F_p(|\vec{q}|) = \int \mathrm{d}^3 r\, e^{i\vec{q}\cdot\vec{r}} \rho_p(r) \ , \tag{9}$$

in the rest frame of the ion; in this case we have $\vec{q}^2 = Q^2$, so that written covariantly this corresponds to the $F_E(Q^2)$ which appears in (6).

Now, as usual we must also account for the so-called survival factor, that is the probability of no additional inelastic hadron–hadron interactions, which would spoil the required exclusivity of the event. This is discussed in [1], and we only briefly highlight the relevant elements here. To account for these effects, we do not apply (1) directly, but rather work at the amplitude level. Focussing on the dominant contribution from the electric from factor, $F_E$, we write

$$T(q_{1\perp}, q_{2\perp}) = \mathcal{N}_1 \mathcal{N}_2 \, q_{1\perp}^\mu q_{2\perp}^\nu V_{\mu\nu} \, , \tag{10}$$

where $V_{\mu\nu}$ is the $\gamma\gamma \to X$ vertex, and the normalization factors are

$$\mathcal{N}_i = \left( \frac{\alpha}{\pi x_i} (1 - x_i) F_E(Q_i^2) \right)^{1/2} \frac{1}{q_{i\perp}^2 + x_i^2 m_{N_i}^2} \, . \tag{11}$$

In terms of this, the production cross section (1) is given by

$$\sigma_{N_1 N_2 \to N_1 X N_2} = \int dx_1 dx_2 d^2 q_{1\perp} d^2 q_{2\perp} \mathcal{PS}_i |T(q_{1\perp}, q_{2\perp})|^2 \, , \tag{12}$$

where $\mathcal{PS}_i$ is defined for the $2 \to i$ process to reproduce the corresponding cross section $\hat{\sigma}$, i.e. explicitly

$$\mathcal{PS}_1 = \frac{\pi}{M_X^2} \delta(\hat{s} - M^2) \, , \qquad \mathcal{PS}_2 = \frac{1}{64\pi^2 M_X^2} \int d\Omega \, . \tag{13}$$

One can show that in the kinematic regime relevant to the EPA, (12) reduces to (1). However, by working with the amplitude $T$ directly we can readily account for soft survival effects. We again refer the reader to [1] for details of this, but simply note here that this is most straightforwardly expressed in impact parameter space, where the average survival factor is given by

$$\langle S_{\text{eik}}^2 \rangle = \frac{\int d^2 b_{1\perp} \, d^2 b_{2\perp} \, |\tilde{T}(s, b_{1\perp}, b_{2\perp})|^2 \exp(-\Omega_{N_1 N_2}(s, b_\perp))}{\int d^2 b_{1\perp} d^2 b_{2\perp} \, |\tilde{T}(s, b_{1\perp}, b_{2\perp})|^2} \, , \tag{14}$$

where $b_{i\perp} = |\vec{b}_{i\perp}|$ is the impact parameter vector of ion $i$, so that $\vec{b}_\perp = \vec{b}_{1\perp} + \vec{b}_{2\perp}$ corresponds to the transverse separation between the colliding ions. $\tilde{T}(s, b_{1\perp}, b_{2\perp})$ is the amplitude (10) in impact parameter space, and $\Omega_{N_1 N_2}(s, b_\perp)$ is the ion–ion opacity; physically $\exp(-\Omega_{N_1 N_2}(s, b_\perp))$ represents the probability that no inelastic scattering occurs at impact parameter $b_\perp$.

We note that in (14) the $\gamma\gamma \to X$ (with $X = l^+ l^-$ in the current case) amplitude has an impact parameter dependence, which we correctly account for in our approach. This derives from the dependence in momentum space of the amplitude on the transverse momenta $q_{i\perp}$ of the incoming photons, which itself is driven by the helicity structure of the corresponding amplitudes (recalling in particular that the photon polarization vector $\epsilon(q) \propto q_{i\perp}$ in the on–shell limit). This modifies both the value of the survival factor, and leads to a process dependence in it. This is often ignored in the literature, see e.g. [6, 28, 29], but we emphasise is a physical effect that should be included.

## 2.2 Removing the $b_{i\perp} < R_A$ region

The $\exp(-\Omega_{N_1 N_2}(s, b_\perp))$ factor in (14) is approximately given by

$$e^{-\Omega(s, b_\perp)/2} \approx \theta(b_\perp - 2R_A) \, , \tag{15}$$

that is, it strongly damps the cross section for hadron–hadron impact parameters less than $2R_A$, where the probability of additional inelastic interactions is rather high; though we emphasise

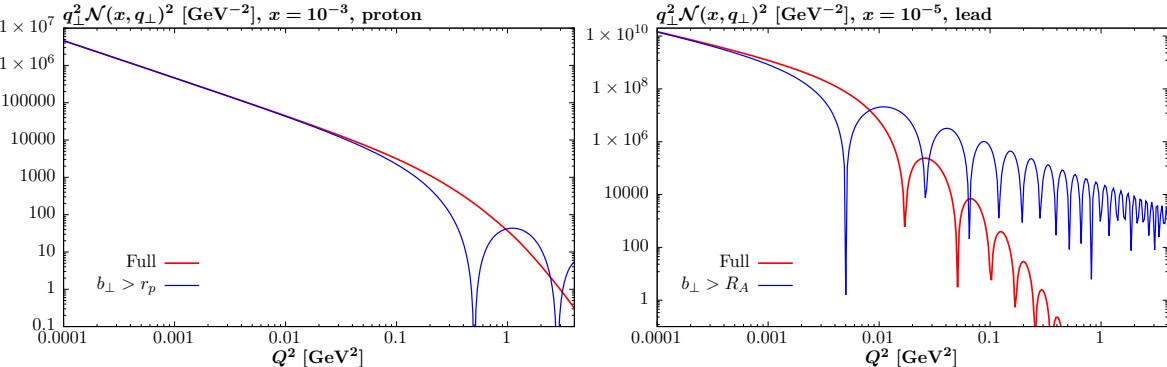

**Figure 1:** Comparison of $q_\perp^2 \mathcal{N}(x, q_\perp)^2$ with and without the cut $b_{i\perp} > R_A$ imposed, as described in the text. The proton (lead) case is shown in the left (right) plots, and representative values of $x = 10^{-3}$ ($10^{-5}$) are taken, corresponding to the production of $\sim 10$ GeV system at central rapidity for $\sqrt{s} = 13$ TeV ($\sqrt{s_{NN}} = 5.02$ TeV).

that in our calculation we give a more complete treatment of the opacity, which accounts for the matter distribution within the hadrons as well as the QCD interaction probability and range. Nonetheless, to first approximation this therefore corresponds to simply limiting the $b_{i\perp}$ integral in (14) so that $|\vec{b}_{1\perp} + \vec{b}_{2\perp}| > 2R_A$. In addition to this, in various places in the literature a further cut is placed on the *individual* impact parameters

$$b_{1,2\perp} > R_A ,\tag{16}$$

between the hadrons and the produced system $X$. See e.g. [6, 28] in the context of $pp$ collisions, and in particular the `STARlight` MC generator [29]. The motivation for this cut is that the final state itself may otherwise interact with the hadron, spoiling the exclusivity of the event. While potentially relevant for the production of strongly interacting states, this is certainly not the case for lepton pairs, see [1, 30–32] for discussion. In particular, such a cut effectively assumes the lepton pair can interact strongly with the hadrons, which is certainly not true. In principle additional QED exchanges between the lepton pair and the ions can play a role, but the impact of this higher order QED effect should not be accounted for according to the above procedure, as in particular this is a higher order QED effect that will not be localised in the $b_{1,2\perp} < R_A$ region, given the long range nature of QED, and nor would it be expected to lead to inelastic production with unity probability in this region, as such a cut implies. We discuss this further in Section 4.2, but the impact of such higher order corrections is expected to be small.

To assess the impact of this cut, we can simply remove the corresponding $b_{i\perp} < R_A$ region from the hadron form factor, in impact parameter space. In more detail, we define

$$F^\mu(x_i, q_{i\perp}) = q_{i\perp}^\mu \mathcal{N}_i(x_i, q_{i\perp}) ,\tag{17}$$

where we explicitly include the $q_\perp$ and $x$ arguments for clarity. We will in particular focus purely on the dominant $\sim F_E$ component of the cross section, as this is sufficient to demonstrate the impact of such a cut. In this way we have

$$T(q_{1\perp}, q_{2\perp}) = F^\mu(x_1, q_{1\perp}) F^\nu(x_2, q_{2\perp}) V_{\mu\nu} ,\tag{18}$$

as in (10), and the cross section follows as before. We then define

$$\tilde{F}^\mu(x_i, b_{i\perp}) = b_{i\perp}^\mu \tilde{\mathcal{N}}_i(x_i, b_{i\perp}) ,\tag{19}$$

as the Fourier conjugate of (17), i.e. so that

$$\tilde{\mathcal{N}}_i(x_i, b_{i\perp}) = \frac{1}{|\vec{b}_{i\perp}|^2} \frac{1}{(2\pi)^2} \int \mathrm{d}^2 q_{i\perp} \, \vec{b}_{i\perp} \cdot \vec{q}_{i\perp} \mathcal{N}_i(x_i, q_{i\perp}) \, e^{i\vec{b}_{i\perp} \cdot \vec{q}_{i\perp}} .\tag{20}$$

We can then define

$$\mathcal{N}_i^{b_{i\perp}<R_A}(x_i,q_{i\perp}) = \frac{1}{|\vec{q}_{i\perp}|^2}\int \mathrm{d}^2 b_{i\perp}\,\vec{q}_{i\perp}\cdot\vec{b}_{i\perp}\tilde{\mathcal{N}}_i(x_i,b_{i\perp})\,e^{-i\vec{b}_{i\perp}\cdot\vec{q}_{i\perp}}\theta(R_A-b_{i\perp})\,, \qquad (21)$$

which in the $R_A\to\infty$ limit simply reproduces the original $\mathcal{N}_i(x_i,q_{i\perp})$. Then, to include the effect of this cut we simply replace

$$\mathcal{N}_i(x_i,q_{i\perp}) \to \mathcal{N}_i^{b_{i\perp}>R_A}(x_i,q_{i\perp}) \equiv \mathcal{N}_i(x_i,q_{i\perp}) - \mathcal{N}_i^{b_{i\perp}<R_A}(x_i,q_{i\perp})\,. \qquad (22)$$

We note that in principle one could of course simply work with $\mathcal{N}_i^{b_{i\perp}>R_A}(x_i,q_{i\perp})$ directly by imposing this condition in (21), but in that case one runs into issues with the numerical stability of the resulting Fourier transform.

The result of imposing this cut is shown in Fig. 1, along with the default case for comparison, with the proton (lead) cases shown in the left (right) plots. For the lead ion, here and in what follows we take $R_A = 6.68$ fm and $d = 0.447$ fm, as given in [37] for the Pb form factor. For the evaluation of survival effects, the neutron density is also required (see [26] for details), for which we take the same Wood–Saxons distribution, but with $R_n = 6.67$ fm and $d_n = 0.55$ fm, again from [37]. For the proton case, as mentioned above we take a fit to the A1 collaboration [35] for the proton form factor. When imposing the $b_{i\perp} > R_A$ cut we take the same value for the Pb case, while to be consistent with [28] in the proton case we take the two dimensional radius, $r_p = 0.64$ fm, determined in the transverse plane, as measured by H1 [38].

We can see that at sufficiently low $Q^2$ the two results coincide, as we would expect given this will be dominated by the higher $b_{i\perp}$ region in impact parameter space, where the cut will have no impact. On the other hand, as $Q^2$ increases we can see that the $b_{i\perp} > R_A$ cut begins to suppress the corresponding result. This is in particular begins to occur for $Q^2 \sim 1/R_A^2$, which is $\sim 0.1$ $(10^{-3})$ GeV$^2$ in the proton (lead) case, as we would expect. As $Q^2$ increases further, we begin to see a dip pattern emerging, due to the fact that the sign of $\mathcal{N}(b_{i\perp} > R_A)$ is changing (for the original $\mathcal{N}$ in the lead case this is due to the Fourier transform (9) that determines the form factor). The magnitude of this in particular becomes larger than the original $\mathcal{N}$ is some regions of $Q^2$, in particular in the lead case. This effect is due to the modulating sign in the Fourier transform (21) and the equivalent expression without the $b_{i\perp} < R_A$ cut, which corresponds to the full $\mathcal{N}$ case. This may appear at first to be counterintuitive, given we are explicitly removing a contribution from the $b_{i\perp} < R_A$ region, but the only requirement this gives is that cross section integrated over $b_{i\perp}$, or equivalently $q_\perp$ in transverse momentum space, is reduced after we impose this cut. Explicitly integrating over the form factors, we observe that this is indeed the case, which the first dip at $Q^2 \sim 1/R_A^2$ providing the dominant impact, while the following peaks occur in rather suppressed regions of phase space. We will confirm this explicitly in the sections which follow. We note that if we instead impose a somewhat smoother requirement than the sharp cutoff $b_{i\perp} < R_A$, then this peaking is somewhat reduced, though not removed entirely.

## 3   Results

### 3.1   Ultraperipheral PbPb collisions: comparison to ATLAS data

We first consider the case of lepton pair production in ultraperipheral heavy ion collisions. Specifically, we compare to the recent ATLAS measurement [24] of muon pair production at $\sqrt{s_{NN}} = 5.02$ TeV in PbPb collisions. Here, a fiducial cross section of $\sigma_{\mathrm{fid.}}^{\mu\mu} = 34.1\pm0.4\,(\mathrm{stat.})\pm 0.7\,(\mathrm{syst.})\,\mu$b is reported. This is compared with the STARlight MC prediction [39] of 32.1 $\mu$b, which is a little lower than the data, and indeed once this is interfaced to PYTHIA8 for QED FSR

| | ATLAS data [24] | Pure EPA | $b_{i\perp} > R_A$ | $b_{i\perp} > R_A$, inc. $S^2$ | inc. $S^2$ | inc. $S^2$ + FSR |
|---|---|---|---|---|---|---|
| $\sigma$ [$\mu$b] | $34.1 \pm 0.8$ | 52.2 | 37.1 | 29.9 | 38.9 | 37.3 |

**Table 1:** Comparison of predictions for exclusive dimuon production in ultraperipheral PbPb collisions, with the ATLAS data [24] at $\sqrt{s_{NN}} = 5.02$ TeV. The muons are required to have $p_\perp^\mu > 4$ GeV, $|\eta^\mu| < 2.4$, $m_{\mu\mu} > 10$ GeV, $p_\perp^{\mu\mu} < 2$ GeV. The data uncertainties correspond to the sum in quadrature of the statistical and systematic.

from the leptons the prediction drops further to 30.8 $\mu$b; given such FSR effects are certainly present this is therefore the more appropriate number for comparison.

We recall from the discussion above, that `STARlight` imposes precisely the $b_{i\perp} > R_A$ cut described in Section 2.2. It is therefore interesting to investigate the impact of this cut on the predicted cross section. In Table 1 we show results for this, as given by `SuperChic 4` [1], suitably modified to include the $b_{i\perp} > R_A$ cut when required. Excluding survival effects, we can see that the impact of this cut is rather significant, reducing the cross section by $\sim 30\%$. A further reduction of a little over $\sim 10\%$ is then introduced by including the physical effect of the survival factor. The final result of 29.9 $\mu$b is a little lower than, but comparable to, the `STARlight` prediction of 32.1 $\mu$b. We note that we do not expect the results to coincide precisely, as e.g. our treatment of survival effects is more complete. In particular, as discussed above we fully account for the impact parameter dependence of the $\gamma\gamma \to \mu^+\mu^-$ amplitude, which is not included in [39]. Nonetheless, we can see that the agreement is significantly improved once the $b_{i\perp} > R_A$ cut is imposed in the `SuperChic` results.

If we exclude this cut, then the survival factor reduces the cross section by $\sim 25\%$, and the resulting cross section is 38.9 $\mu$b, i.e. is as expected higher. Thus, we can indeed confirm the fact that it is only by including this unphysical cut that consistency with `STARlight` is found. Now, our baseline prediction of 38.9 $\mu$b lies above the data, though we should bear in mind that the impact of QED FSR is found in the analysis to reduce the `STARlight` prediction by $\sim 4\%$, and so will be expected to reduce our prediction to $\sim 37.3$ $\mu$b; this is given in the last column of Table 1 for comparison. This is still in rather poor agreement with the data, lying above it, though the `STARlight` predictions undershoot the data by a similar amount.

We now consider the impact on the differential predictions. It was in particular observed in [24] that the `STARlight` predictions tend to undershoot the data as the dimuon rapidity, $|y_{\mu\mu}|$, is increased. Given the discussion above, it is interesting to examine whether the imposition of the $b_{i\perp} > R_A$ cut, as well as modifying the total cross section, might modify the resulting rapidity distribution in such a way as to explain this discrepancy. We therefore plot in Fig. 2 (top left) the ratio of the normalized distribution using our default ('full') prediction to that found by imposing the $b_{i\perp} > R_A$ cut. We consider the normalized case in order to isolate the impact on the shape alone. We can clearly see that the effect is rather large, with the cut leading to a decrease in the normalized distribution at higher rapidities by $\sim 15\%$. Crucially, we can see from Fig. 6 of [24] that the shape and magnitude of the trend closely follows that observed when plotting the ratio of the data to the `STARlight` prediction. That is, this is undershooting the data by precisely the level we would expect from Fig. 2 (top left), given that the $b_{i\perp} > R_A$ cut is being imposed. Removing this artificial cut will therefore clearly lead to a better description of the rapidity distribution.

In [24] a related effect is also seen with respect to the minimum and maximum photon energies, defined via the minimum/maximum value of $k_{1,2} = \sqrt{s}x_{1,2}/2$, where $x_{1,2}$ are the photon momentum fractions. Here, the `STARlight` predictions are observed to undershoot the data at both lower and higher values of $k_{min}$ and $k_{max}$. In Fig. 2 (top right) we plot the same ratio of normalized distributions as before, but now with respect to these variables. Remarkably, comparing with Fig. 10 of [24] we can see that precisely this trend is reproduced by our results,

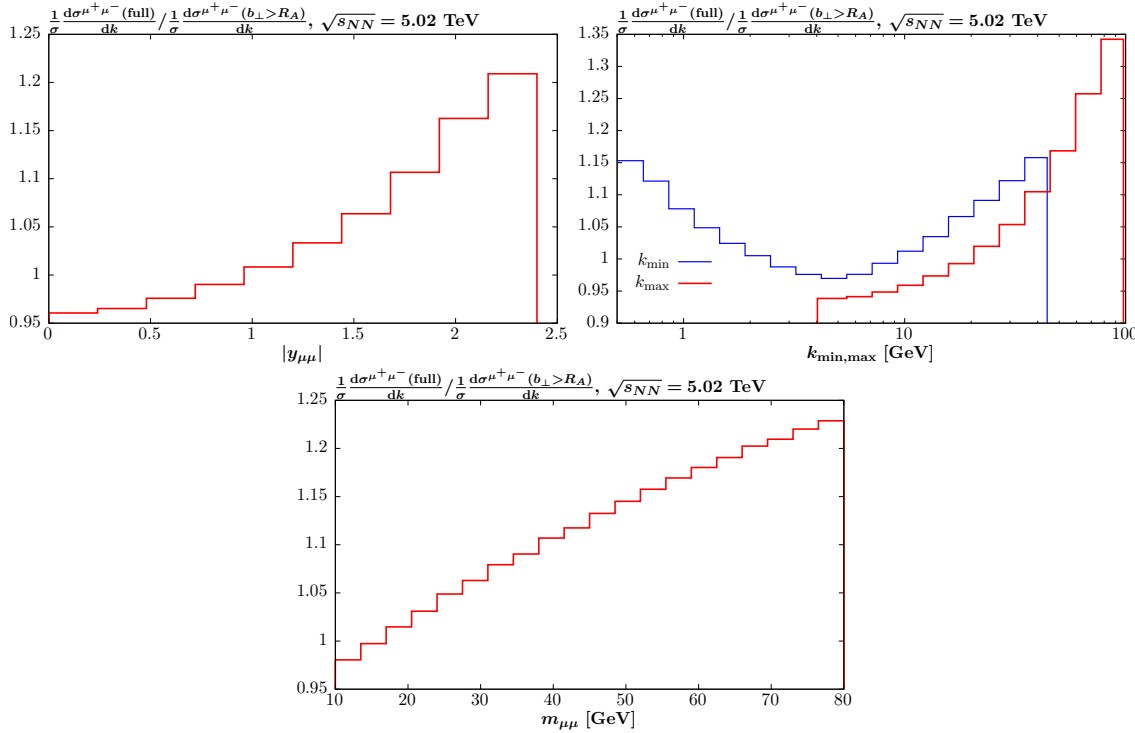

**Figure 2:** Normalized differential cross sections as a function of the (top left) dimuon rapidity, (top right) maximum photon energy ($k_{\mathrm{max}}$) and minimum photon energy ($k_{\mathrm{min}}$), and (bottom) dimuon invariant mass, calculated using a modified version of `SuperChic 4` [1]. The ratio of the full result to the case with the $b_{i\perp} > R_A$ cut imposed is given; in both cases the survival factor is included.

and hence once again we can expect a greatly improved description of these distributions by removing the $b_{i\perp} > R_A$ cut. This distribution in addition gives some insight into the reason why this cut affects the results differentially in such a way. In particular, we can see from (3) that the minimum value of the photon $Q_i^2$ is proportional to the momentum fraction $x_i^2$. Higher values of $k_{\mathrm{max}}$ correspond to higher values of the corresponding photon momentum fraction, and hence higher values of $Q_i^2$ on average. We can then see from Fig. 1 that larger $Q_i^2$ is precisely where the impact of the $b_{i\perp} > R_A$ cut is higher; in particular as the interaction is then less peripheral. This effect in addition explains the impact of the cut on higher rapidities, which are correlated with an increased $k_{\mathrm{max}}$. While the corresponding $x_i$ value of the other photon in this case will be lower, and hence one would expect a reduced impact from the cut on this side, it is clear from our results that it is the effect of increasing $x_i$ that dominates.

The enhancement in the low $k_{\mathrm{min}}$ case is therefore simply because this is kinematically correlated with larger $k_{\mathrm{max}}$ for the other photon. In particular, for $y_{\mu\mu} = 0$ we have $k_{\mathrm{min}} = 5$ GeV, due to the lower limit on $m_{\mu\mu}$ in the data, and hence indeed the region of $k_{\mathrm{min}}$ below this is due to production away from central rapidities. The enhancement for $k_{\mathrm{min}}$ values above this corresponds to the larger $m_{\mu\mu}$ region, which are rather kinematically suppressed. Nonetheless, again in [24] there is some hint of a corresponding excess in the ratio of data to `STARlight`, albeit within very limited statistics.

A further way we can examine the effect of this cut is to consider the invariant mass distribution, which is shown in Fig. 2 (bottom). We can see that here the $b_{i\perp} > R_A$ cut reduces the cross section more significantly at higher masses, precisely in line with the discussion above, as this will correspond to larger photon $x_i$ values on both sides. Interestingly, in Fig. 7 of [24] there is no clear sign of any deviations with respect to `STARlight` predictions in the ATLAS

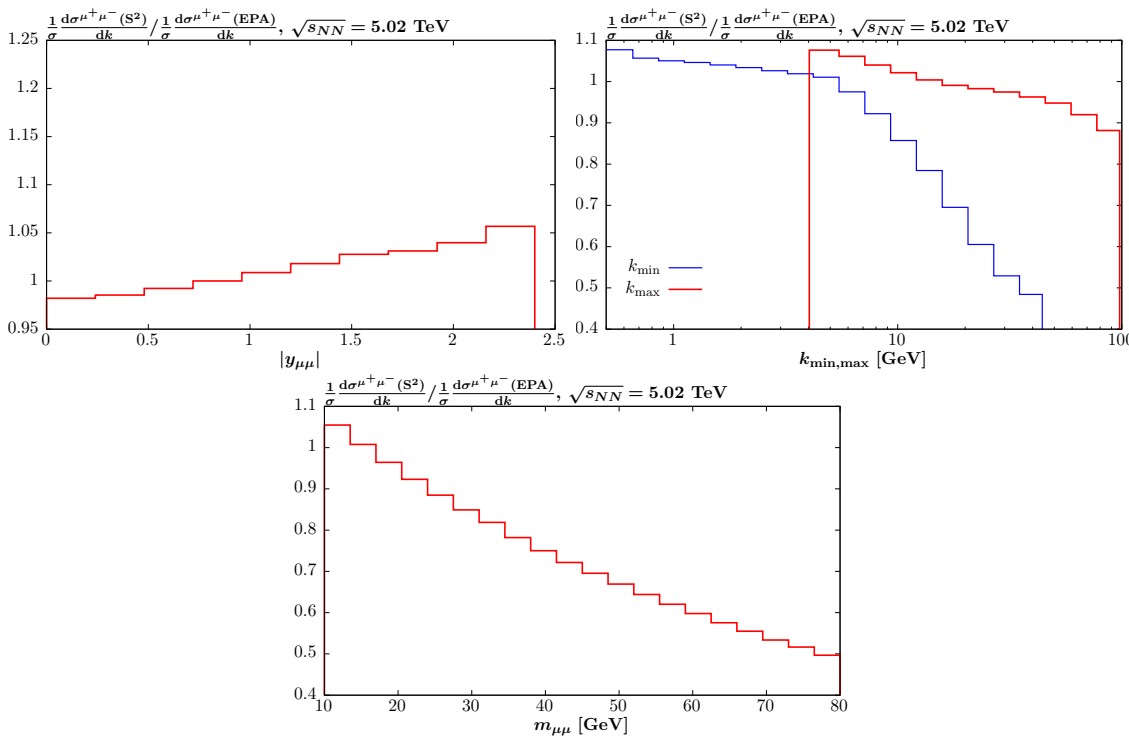

**Figure 3:** Normalized differential cross sections as a function of the (top left) dimuon rapidity, (top right) maximum photon energy ($k_{max}$) and minimum photon energy ($k_{min}$), and (bottom) dimuon invariant mass, calculated using a modified version of `SuperChic 4` [1]. The ratio of the full result including the survival factor to the EPA result, i.e. excluding this, is shown; in both cases no $b_{i\perp} > R_A$ cut is imposed. We emphasise that in the corresponding absolute distributions the results including survival effects will be suppressed with respect to the pure EPA.

data, however here the statistics become rather limited above the $m_{\mu\mu} \sim 40$ GeV region, which we can see from Fig. 2 (bottom) is where the difference is largest. We would certainly expect to see this trend confirmed in future data.

Finally, in Fig. 3 we show results for the same normalized distributions as before, but now considering the ratio of the predictions including the survival factor to that excluding it. Here, this physical effect must certainly be included, and it is interesting to study the impact this has on the distributions, in addition to the overall reduction in rate that it leads to. We emphasise that by plotting the normalized distributions the impact that survival effects have in reducing the overall rate is factored out, and we can instead focus on its effect on the shape of differential observables. The impact on the rapidity distribution is quite a bit milder than in the case of the $b_{i\perp} > R_A$ cut, and overall tends to increase the relative contribution to the cross section at larger rapidities. Interestingly, the opposite trend is observed in [1] for the case $pp$ collisions, i.e. the predicted survival factor decreases at larger rapidities. Again, for larger average $Q^2$ we probe on average smaller impact parameters and so the impact of survival effects will become larger. However as noted above, for forward rapidities we probe higher values of $x_i$ on one photon side, but lower values on the other, and hence it is difficult from first principles to predict what the trend will be. In particular, this should depend on the specific $Q^2$ distribution of the hadron form factors, and indeed we can see that this is the case here, giving the differing trends in the proton and lead cases. It is also of note that in the proton case, we predict in [1] an increase in the survival factor at forward rapidities for inelastic photon emission from both protons, i.e. double dissociative production. In the $k_{min,max}$ distributions we can see that the survival

|  | ATLAS data [14, 16] | Pure EPA | inc. $S^2$ | $b_{i\perp} > r_p$ | $b_{i\perp} > r_p$, inc. $S^2$ |
|---|---|---|---|---|---|
| $\sigma$ [pb], 7 TeV | $0.628 \pm 0.038$ | 0.798 | 0.742 | 0.660 | 0.626 |
| $\sigma$ [pb], 13 TeV | $3.12 \pm 0.16$ | 3.58 | 3.43 | 3.12 | 3.02 |

**Table 2:** Comparison of predictions for exclusive dimuon production in $pp$ collisions, with the ATLAS data [14,16] at $\sqrt{s} = 7$ and 13 TeV, within the fiducial acceptance. The data uncertainties correspond to the sum in quadrature of the statistical and systematic.

factor is smaller for larger values, and again at larger invariant masses a sizeable suppression is observed. This is due to the same effect as that discussed above, namely that at higher masses the cross section probes larger values of $x_i$ for both photons and hence the reaction tends to be less peripheral.

### 3.2   $pp$ collisions

We now consider exclusive PI production in $pp$ collisions. We compare to the ATLAS data [14,16] at 7 and 13 TeV, which are collected without tagged protons and corrected experimentally back to a purely elastic cross section. We do not compare to the more recent ATLAS data with a single proton tag [10], as although this in principle corresponds to a cleaner data sample, the experimental errors are rather larger. A $b_{i\perp} > r_p$ cut is imposed in the predictions of [28], which are compared to ATLAS data in [14,16], at 7 and 13 TeV. In the 7 (13) TeV case the muon pair invariant mass is restricted to be $m_{\mu\mu} > 20$ (12) GeV, with further cuts imposed as described in the corresponding references. Cross section results are shown in Table 2, in the same format as Table 1. We can see that in both cases the impact of imposing the $b_{i\perp} > r_p$ cut, which reduces the 7 (13) TeV cross section by $\sim 17\%$ (13%), is rather larger than the impact of the survival factor, which reduces it by $\sim 7\%$ (4%). Moreover, we can see that the predicted value for the cross sections including both the $b_{i\perp} > r_p$ cut and survival effects is rather close to those quoted in [14, 16], corresponding to the predictions of [28]. For example, in the 13 TeV case a central prediction of 3.06 pb is quoted, which is very close to our result of 3.02 pb. As in the comparison to `STARlight` in the PbPb case, we do not expect our results to coincide exactly, due to the fact that we account for the impact parameter dependence of the $\gamma\gamma \to \mu^+\mu^-$ amplitude, and indeed we take a more precise fit to the proton form factor. Nonetheless, we can see that our results agree rather well once the $b_{i\perp} > r_p$ cut is imposed in the `SuperChic` results.

## 4   What are the theoretical uncertainties?

In the previous sections, we have seen that without the artificial $b_{i\perp} > r_p$ cut, our predictions in $pp$ collisions lie $\sim 2 - 3\sigma$ above the data, while for PbPb our result lies $\sim 4\sigma$ above the data. Given this, it is natural to investigate possible causes for such an excess in the theoretical calculation. These comparisons only account for experimental uncertainties, and hence as a first step we should evaluate the corresponding theoretical uncertainties. As we will see, these are in general expected to be very small; to emphasise this point we will consider in some cases rather extreme variations in the model parameters that are physically disfavoured but even then lead to rather small changes in the predicted cross sections.

### 4.1   $pp$ collisions

We begin with the case of $pp$ collisions. A first natural source of uncertainty to consider is in the input elastic proton form factors, which as described in Section 2.1 are taken from a fit

|  | ATLAS data [14, 16] | Baseline | FF uncertainty | Dipole FF |
|---|---|---|---|---|
| $\sigma$ [pb], 7 TeV | $0.628 \pm 0.038$ | 0.742 | $^{+0.003}_{-0.005}$ | 0.755 |
| $\sigma$ [pb], 13 TeV | $3.12 \pm 0.16$ | 3.43 | $\pm 0.01$ | 3.48 |

**Table 3:** Comparison of predictions for exclusive dimuon production in $pp$ collisions, as in Table 2, but showing the uncertainty in the theoretical predictions due to the proton form factors (FFs), evaluated as described in the text. Also shown, for comparison, is the result using the dipole form factor (5). All results include the survival factor.

due to the A1 collaboration [35]. To evaluate the uncertainty on this, we add in quadrature the experimental uncertainty on the polarized extraction and the difference between the unpolarized and polarized cases. This gives an uncertainty on the form factors $G_{E,M}$ that is at the sub–percent level in the lower $Q^2$ region relevant to our considerations. We show in Table 3 the impact of this on the same $pp$ cross sections as before, and can see that they are less than 1% and hence are under good control. As an aside, we also show results with the rather approximate dipole form factor (5). Here the difference is a little larger, though still rather small. Thus even taking this rather approximate and extreme case (the dipole form factor is certainly disfavoured experimentally) leads to very little difference in the result. In other words, this is a negligible source of uncertainty with respect to the measurements we consider here.

We next consider the uncertainty due the survival factor. We can see that this reduces the predicted cross sections by $\sim 7$ (4) % in the 7 (13) TeV cases, with the difference being primarily driven by the lower dimuon invariant mass cut in the 13 TeV case. These are clearly rather mild suppressions, which as discussed in e.g. [1,25] are driven by the peripheral nature of photon–initiated process. In particular, the elastic proton form factors are strongly peaked at low photon $Q^2$, and in impact parameter space this corresponds to rather large proton–proton impact parameters, $b_\perp$.

Nonetheless, one might then wonder if a different modelling of such effects could reasonably lead to a somewhat larger suppression, and hence a better matching of the data. As a first attempt, we could consider taking the different models described in [33], which all correspond to two–channel eikonal models that provide an equally good description of the available hadronic data at the time, but with rather different underlying parameters. The difference between these is in general rather large, and in this study it is shown that the predicted survival factor for exclusive SM Higgs Boson production varies by a factor of $\sim 3$ between the different models; for such a QCD–initiated process the reaction is significantly less peripheral and therefore the dependence on the model of the survival factor correspondingly larger. Taking these alternative models (we take model 4 for concreteness in our baseline predictions) in the current case, however, we find the variation is negligible, at the per mille level.

To investigate this effect further, we consider some more dramatic (and certainly experimentally disfavoured) variations in the modelling of the survival factor. We in particular consider a simplified 'one–channel' model, as in e.g. [40]. That is, we ignore the internal structure of the proton, and assume the proton–proton elastic scattering amplitude is given by a single Pomeron exchange, with

$$A_{pp}(s, k_\perp^2) = i s C^* \sigma_{\text{pp}}^{\text{tot}}(s) \exp\left(-B k_\perp^2 / 2\right) . \tag{23}$$

The proton opacity $\Omega_{pp}(s, b_\perp)$ appearing in (14) is given in terms of the Fourier transform of this, i.e.

$$\Omega_{pp}(s, b_\perp) = \int \text{d}^2 k_\perp \, e^{-i \vec{b}_\perp \cdot \vec{k}_\perp} A_{pp}(s, k_\perp^2) . \tag{24}$$

Here taking $C^* \neq 1$ physically provides an effective way of accounting for the possibility of proton excitations ($p \to N^*$) in the intermediate states. As discussed in [40], a value of $C^* \sim 1.3$ gives

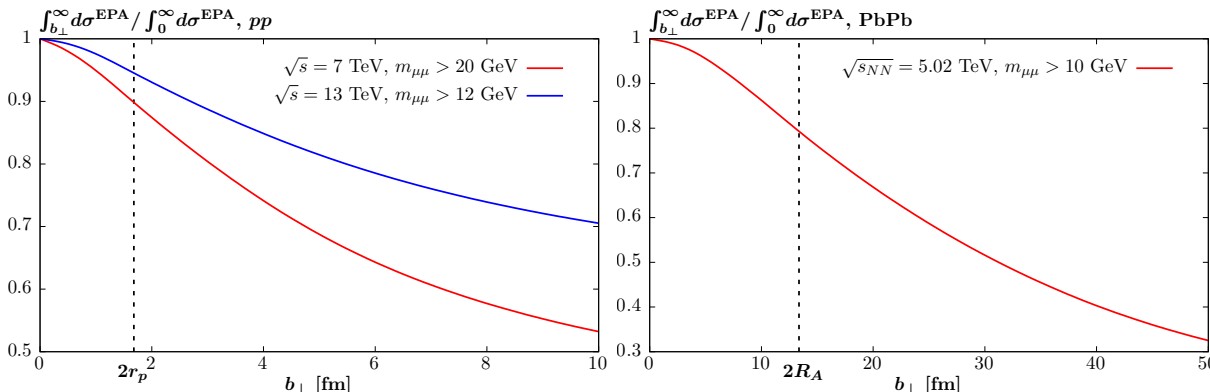

**Figure 4:** The pure EPA predictions for the ATLAS $pp$ [14, 16] and PbPb [24] data as a function of a lower cut on the hadron–hadron impact parameter $b_\perp$, considered as a ratio to the full EPA result, i.e. integrated down to zero $b_\perp$. All results apply the corresponding experimental event selection. The values of twice the proton and lead radii are indicated.

a similar value for the survival factor to the more complete two–channel approach. However, for our purposes we do not pursue this interpretation further, but simply treat this as a free parameter with which to investigate the impact of modifications to the description of proton–proton interactions on the survival factor. We can in effect interpret variations of $C^*$ about this value as corresponding variations in the input value of the $\sigma_{pp}^{tot}$, which is known experimentally with percent level precision. Such an interpretation is not completely direct, as in reality a more complete modelling is required than this single–channel approach, but it allows us to get a handle on how quite extreme variations in this parameter give rather small effects on the survival factor.

In Table 4 we show results for 7 and 13 TeV as before, but using the above simplified model of the survival factor, and consider a very extreme range of $C^* = 1-2$. We emphasise that such a range is certainly incompatible with existing data on hadronic interactions, e.g. the upper (lower) end will correspond to values of $\sigma_{pp}^{tot}$ that are far too high (low). However, even taking this extreme range we can see that the corresponding variation in the survival factor is relatively small, with the lower end of the predictions (corresponding to $C^* = 2$) still overshooting the ATLAS data. This result is indicative of a straightforward geometric fact about the elastic photon–initiated cross section, namely that even taking an artificially large inelastic proton–proton scattering cross section, there is a sizeable fraction of the cross section that in impact parameter space is simply outside the range of such inelastic QCD interactions.

To demonstrate this, in Fig. 4 we show the pure EPA predictions for the ATLAS $pp$ and PbPb data as a function of a lower cut on the hadron–hadron impact parameter $b_\perp$, considered as a ratio to the full EPA result, i.e. integrated down to zero $b_\perp$. This shows the fractional contribution to the total cross sections, prior to including survival effects, coming from the region of impact parameter space greater than a given $b_\perp$, and is therefore a measure of precisely how peripheral the interaction is. We can see that in all cases a significant fraction of the cross section comes from the region of rather high $b_\perp \gg 2r_p, 2R_A$, which we can therefore expect to be untouched by survival effects, irrespective of the particular model applied. We note that the difference between the 7 and 13 TeV $pp$ cases is driven primarily not by the c.m.s. energy but rather the lower $p_\perp$ cut in the 13 measurement, which as discussed above leads to a more peripheral interaction; this is clearly seen in the figure. Due to the larger ion radius, the PbPb is as expected significantly more peripheral, though the impact of survival effects will of course extend out to much larger $b_\perp$ for the same reason.

|  | ATLAS data [14, 16] | 1 ch. $(C^* = 1 - 2)$ | $\theta(b_\perp - 2r_p)$ | $\theta(b_\perp - 3r_p)$ |
|---|---|---|---|---|
| $\sigma$ [pb], 7 TeV | $0.628 \pm 0.038$ | 0.748 - 0.727 | 0.719 | 0.668 |
| $\sigma$ [pb], 13 TeV | $3.12 \pm 0.16$ | 3.45 - 3.40 | 3.34 | 3.25 |

**Table 4:** Comparison of predictions for exclusive dimuon production in $pp$ collisions, as in Table 2, but considering extreme variations in the modelling of survival effects, as described in the text.

Now, we recall from (15) that the survival factor can be approximated by assuming that the hadrons will interact inelastically with unit probability provide they overlap in impact parameter, that is taking

$$e^{-\Omega(s, b_\perp)/2} \approx \theta(b_\perp - 2r_p). \tag{25}$$

The impact of this can be read off from Fig. 4, and is shown in Table 4, taking $r_p = 0.84$ fm. We can see that this already rather extreme assumption leads to a somewhat lower survival factor, though still giving a cross section that lies above the data. As an exercise, we can then consider taking

$$e^{-\Omega(s, b_\perp)/2} \approx \theta(b_\perp - 3r_p), \tag{26}$$

i.e. assuming that the inelastic scattering probability is unity if the proton edges are separated by $r_p$ or less. With this level of highly unphysical behaviour we finally find results that are more consistent with the data. This brings the issue into rather stark focus: the only way we can account for the overshooting of the data here, if we are to only modify the modelling of survival effects in the elastic case, would be to take an approach that roughly corresponds to the level of suppression given by (26), or even higher. This is certainly ruled out by basic observations about the range and strength of proton–proton QCD interactions.

Finally, we note that the focus of this discussion has been on purely elastic production, given the most precise ATLAS data on this [14, 16] are provided as cross sections corrected back to a purely exclusive result. However, in general the initiating photons can be emitted inelastically from the protons, see [1] for a detailed discussion. We may therefore ask how the theoretical uncertainties are affected in such a case. First, in terms of the proton form factors, these can be expected to have a somewhat larger uncertainty, as these are somewhat less well constrained than for purely elastic scattering. Nonetheless, they remain rather well constrained, and the uncertainty associated with this is small. In terms of the survival factor, for single proton dissociation (see [1]) the production process is also highly peripheral, due to the fact that an elastic proton vertex is present on one side. For similar reasons to the elastic case, we therefore expect the model dependence in the survival factor to be rather small, as again a significant fraction of the scattering process will be outside the range of QCD interactions. Nonetheless, the collision is in general less peripheral, and hence there may a somewhat larger theoretical uncertainty in this case. For double dissociative production, the peripheral nature of the interaction is lost, and here the survival factor is significantly lower and indeed more model dependent. However, in general this is found to give a very small contribution to the ATLAS data [14, 16] prior to correcting back to the exclusive case.

Given the discussion above, it is interesting to recall that in [1] a comparison of the `SuperCHIC` predictions for muon pair production differential in the dimuon acoplanarity is compared to the ATLAS 7 TeV data [14], where both the data and theory include elastic and SD production. Here it is was found that the only statistically relevant excess in the theoretical predictions occurs in the lowest acoplanarity bin, which is both where the elastic component is most enhanced and where the interaction most peripheral, i.e. where the value of the survival factor is expected to be largest, and the uncertainty associated with it smallest. It will certainly be of great interest to compare to future more precise data to shed further light on this.

|  | ATLAS data [24] | $\theta(b_\perp - 2R_A)$ | $\theta(b_\perp - 3R_A)$ |
|---|---|---|---|
| $\sigma$ [$\mu$b] | $34.1 \pm 0.8$ | 41.4 | 34.7 |

**Table 5:** Comparison of predictions for exclusive dimuon production in PbPb collisions, as in Table 1, but considering extreme variations in the modelling of survival effects, as described in the text.

## 4.2 PbPb collisions

We next consider the case of heavy ion collisions, again focussing on the comparison to the same ATLAS data as before. A first natural source of uncertainty is again in the electric form factor of the lead ion. To estimate this, we consider a rather extreme variation in the ion radius and/or skin thickness, by $\pm 0.1$ fm for both the neutron and proton cases; we note that the experimental values [37] of these observables are determined with significantly greater precision than this, in particular in the proton case. Even so, this gives at most a $1 - 2\%$ variation in the resulting cross section. The genuine uncertainty from these inputs will therefore be significantly smaller than that.

Next, we consider the impact of survival effects. As discussed in [26], in the heavy ion case these also depend on the modelling of inelastic proton–proton collisions, and as such we could pursue a detailed analysis of model variations in this, as in the proton case. However we have already observed the relative insensitivity to this for proton scattering, and the same will be true here. Therefore, to keep the discussion simple, we just consider the same replacements (25) and (26), but with $r_p \to R_A$. The effect of this is shown in Table 5. We can see that taking (25) gives a slightly larger cross section than our default result of 38.9 $\mu$b: this approximate result misses the finite range of QCD interactions and in particular the non–zero extent of the Pb ion outside $R_A$, and hence underestimates the impact of survival effects somewhat. We can then see that in order to get good agreement with the data by modifying survival effects, we are forced to take a form like (26). Again, this roughly corresponds to the case of unit inelastic scattering probability out to a range of $R_A \approx 6.68$ fm outside the Pb edge. Needless to say, this is physically incompatible with our knowledge of the range and strength of QCD interactions, and hence cannot be the resolution to this discrepancy. In particular, any more realistic model would have to give this level of suppression in order to match the data by modifying the survival factor alone, and hence will be similarly physically ruled out. This is again a result of the peripheral nature of the PbPb collision, as demonstrated in Fig. 4.

We note that there are other potential sources of uncertainty and/or incompleteness in our theoretical description for heavy ion collisions. First, we note that our calculation corresponds to the case of purely elastic emission from the lead ions, whereas the data includes ion dissociation; indeed the fractions with and without this are determined experimentally via measurements with ZDCs in [24]. However, such dissociation is dominantly driven by additional ion–ion photon exchanges. These should occur independently of the lepton pair production process, see [29], and so the total rate is simply given by the prediction for elastic production we present here. That is, the impact of these additional ion–ion photon exchanges is unitary, preserving the overall rate, as calculated for the case of elastic production[1]. In principle this is only true for the integrated cross section, and in particular when cuts on the dimuon $p_\perp$ and/or acoplanarity are imposed,

---

[1]As an aside, we note that `SuperCHIC` predictions are compared against data from the STAR collaboration on lepton pair production in ultraperipheral AuAu collisions in [41]. However, such data correspond to 'tagged' collisions, that is where at least one neutron is required to be emitted from each colliding ion, ensuring that both ions have undergone dissociation. This is in contrast to the ATLAS case, which does not require this, and indeed our approach is not expected to described such tagged data completely, in particular with respect to the lepton pair $p_\perp$ distribution. This point is not expressed clearly in [41], where it is even incorrectly stated that the disagreement observed with `SuperCHIC` invalidates our calculation of purely exclusive production.

this will remove some fraction of the dissociative events in a manner that is not accounted for in our calculation. However, in the ATLAS analysis a reasonably high cut of $p_\perp^{\mu\mu} < 2$ GeV is imposed, which is found to only remove a very small fraction of the `STARlight` predicted events (for which dissociation due to ion–ion photon exchange is included). We note in addition that inelastic production due to emission from the individual protons within the ions (which we do note account for) is subtracted from the data.

However, in addition to the above there are in principle so–called unitary corrections [42, 43], driven by the possibility that further lepton (dominantly electron) pairs can be produced via additional photon–initiated interactions. Due to the $\sim Z^2$ enhancement of the photon flux, the probability for this to occur is rather high, with [43] estimating that $\sim 50\%$ of LHC PbPb muon pair production events will contain at least one additional electron–positron pair, and hence in principle the cross section for producing only one muon pair and nothing else will be correspondingly reduced. However, such additional pair production will be strongly peaked at low dielectron invariant masses and hence these will generally not be expected to fail the experimental veto requirements. Nonetheless, a small fraction may do, due either to the experimental requirement in [24] of no additional reconstructed tracks being present or the minimum–bias trigger scintillator (MBTS) veto that is applied. In the latter case pile-up production of electron pairs may in principle be relevant. A full evaluation of this would require a dedicated study.

A further possibility, again due to the $\sim Z^2$ enhancement of the photon flux, is that there could in principle be a strong impact from higher order QED exchanges between the muon pair and the lead ions, which again is not included in our calculation. The effect of this is however strongly suppressed by a cancellation between the diagrams where the photon is exchanged with the $\mu^+$ and the $\mu^-$, that occurs up to non–zero $\sim Q^2/m_{\mu\mu}^2$ contributions. The final result, according to the specific calculation of [43] is that even at muon threshold, the expected correction is $\sim 1\%$ or less, and hence in the ATLAS case should be negligible. Nonetheless, it is interesting to note that both this effect and the impact of unitary corrections is qualitatively to reduce the theoretical cross section, that is in the direction of the data. We also note that a recent study [32] suggests the impact of these higher order corrections could be at the $\sim 10\%$ level, though this is clearly in contradiction with [43] (see also [44] for a review and further references) and indeed the physical expectations discussed above. We note in addition that such effects are clearly not relevant in $pp$ collisions, where there is no corresponding $\sim Z^2$ enhancement in the photon flux.

Finally, photon FSR from the muon pair certainly plays a role both in $pp$ and PbPb collisions. It is in particular worth emphasising that the experimental selection for these events focusses on the region of very low muon pair acoplanarity. The impact of FSR in this region can be particularly enhanced, generating in particular a Sudakov suppression in the rate as the acoplanarity approaches zero. The impact of FSR in the PbPb case, as modelled via `Pythia` is found to be non–negligible, although here one may expect its effect to be increased by the rather low muon $p_\perp$ threshold, which is higher in the pp measurements. Nonetheless, a more detailed revisiting of the impact of photon FSR may in principle improve the agreement between data and theory, at least somewhat.

## 5   Summary and Outlook

Photon–initiated (PI) production is a unique and highly favourable channel at the LHC, in both $pp$ and heavy ion collisions. This naturally leads to events with intact hadrons and/or rapidity gaps in the final state, which provide a particularly clean experimental and theoretical environment in which to probe the SM and physics beyond it; one can in effect use the LHC as

a photon–photon collider. Indeed, there is an ongoing broad LHC programme of experimental studies of such processes using dedicated proton tagging detectors in association with ATLAS and CMS, as well as in ultraperipheral heavy ion collisions.

A key motivating factor in these studies is that the production mechanism is particularly well understood, in particular in the elastic case, where the hadrons remain intact after the collision. The photon emission probability is given directly in terms of experimentally very well determined hadron EM form factors, while the calculation of the $\gamma\gamma \to X$ subprocess is in general under very good theoretical control, either for the production of SM or indeed BSM states.

However, on top of this one must account for the probability of addition hadron–hadron interactions, which can lead to colour flow between the colliding hadrons and an inelastic event with no intact hadrons or rapidity gaps present. This is naively a significant source of uncertainty, as for a general LHC event the hadron–hadron interaction probability is rather large, and its evaluation rather model dependent. Fortunately though, the elastic PI process in particular is a special case: the emitted photon virtualities are in general low, and hence the impact parameter of the colliding hadrons is in general beyond the range of QCD interactions. Thus, the 'survival factor', $S^2$, i.e. the probability for no additional inelastic hadron–hadron interactions, is rather close to unity.

Nonetheless, the survival factor is not exactly unity, and additional hadron–hadron interactions can and will occur, even if the effect is relatively mild in many cases. It is therefore crucial to have a clear theoretical handle on predictions for this object, and for the uncertainties on it. In this paper we have discussed this question in detail, in particular in light of the fact that the predicted cross section for exclusive PI muon pair production in both $pp$ and PbPb collisions, as implemented in the `SuperCHIC` MC, appear to overshoot the ATLAS data for these processes by $\sim 10\%$.

We have first demonstrated that the dominant reason that the calculations of [28, 29] lie rather lower than our predictions is due not to a genuine model dependence, but rather an unphysical cut that is imposed in these references on the dimuon–hadron impact parameter. Indeed, we have shown that the impact of such a cut is to closely reproduce the discrepancy between the `STARlight` MC results and the muon kinematic distributions measured by ATLAS in PbPb collisions at $\sqrt{s}_{NN} = 5.02$ TeV [24]. Thus it is disfavoured experimentally as well as theoretically. We have demonstrated that once this cut is removed, our results and those of [28, 29] for elastic PI production will be in much better agreement.

Further to this, we have explored the genuine theoretical uncertainty due to the modelling of the survival factor. Considering reasonable model variations within the approach of `SuperChic` we have found that these only effect the predictions at the $\lesssim 1\%$ level, and similarly for uncertainties in the underlying hadron EM form factors. Going further, and considering more extreme, and indeed rather unphysical, variations we have showed that it is only by including a survival probability that corresponds to the case of inelastic hadron–hadron interactions occurring with unit probability out to hadron–hadron impact parameters $b_\perp \sim 3R_A$ that the ATLAS data begins to be matched by the predictions. For PbPb collisions in particular, this separation is well outside the range of QCD. This underlines the basic, rather model independent, point that a significant fraction of elastic PI scattering occurs for hadron–hadron impact parameters that are simply outside the range of QCD interactions, and hence this sets a lower bound on the survival factor in any physically reasonable approach.

Given this, we have also briefly reviewed other potential sources of uncertainty, due to higher order QED effects in PbPb case, and final–state photon emission in both the $pp$ and PbPb cases, but find no clear evidence that these are not under good control. We have demonstrated explicitly for the case of PbPb collisions (and indeed the same remains true for the $pp$ case, see [1]) that

as well as affecting the overall cross section, the survival factor induces distinct modifications to the muon kinematic distributions. We would hope that comparisons of present and in particular future data differentially with the predictions of `SuperChic` could provide evidence for these modifications, and hence of the overall approach.

Beyond this, a closer examination of the role of events with proton dissociation in $pp$ collisions would be worthwhile. The contribution from these is often subtracted in a data–driven way in order to present a purely exclusive cross section, but in the future a comparison with the results of of `SuperChic` for both the elastic and inelastic contributions would be much more direct; see [1] for a first comparison. In particular, we note that while the focus of this article has been on elastic production, for single proton dissociation the production process is also highly peripheral, due to the fact that an elastic proton vertex is present on one side. For similar reasons to those presented in this paper, we therefore expect the model dependence in the survival factor to be rather low. Nonetheless, the collision is in general less peripheral, and hence there may a somewhat larger theoretical uncertainty associated in this case. In this respect, future higher precision updates on the first data on lepton pair production with tagged protons, by both ATLAS and CMS, will we hope shed significant light on these issues. However, for now the source of the apparent data/theory discrepancy remains unclear to us.

## Acknowledgments

We thank Mateusz Dyndal, Peter Steinberg and Marek Tasevsky for useful discussions relating to the ATLAS measurements. We thank Alexander Milstein and Valery Serbo for useful discussions on the role of higher order QED effects in dilepton production in heavy ion collisions. LHL thanks the Science and Technology Facilities Council (STFC) for support via grant award ST/L000377/1.

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
