# Peer review of "Elastic photon-initiated production at the LHC: the role of hadron-hadron interactions"

_SciPost Physics_

## Round 1 · Referee Report · Anonymous (Referee 1) · 2021-5-21

Strengths

-The paper is well written. -We can follow the reasoning of the authors linearly and the attempt of the author to make a proof of one effect is essentially fulfilled. There are three main aspects discussed in the text: -1. The role and significance of the cut b>R in STARlight MC and some other models used for example in ATLAS to predict the cross section of exclusive pairs of muons production. The authors explain why this cut is unphysical while necessary to obtain a correct numerical prediction... The arguments are clearly exposed, with of course the subjectivity inherent to the authors. The authors discuss in particular why the predictions of SuperCHIC may differ from the standard version of STARlight, while by using a similar treatment of the cuts in impact parameter, the predictions tend to converge. -2. How to account for uncertainties for the process here mentioned in a complete ways... For example in the survival factor determination etc. Again, this discussion is clear and can be easily followed. - 3. points 1. and 2. are exposed both in the context of pp and PbPb collisions... Further comments are made in the text to discuss the predictions are very large impact parameter (>3R). - I think that all these items are nice arguments in the field and this discussion is useful in the practical life of experimentalist as well as giving some ideas for mors fundamental views of the topic.

Weaknesses

This is not really a weakness but more an outlook for such studies. We are all aware that these treatments are too simplistic to be correct in fundamental physics. Obviously, wave functions of the incoming particles and the behavior of such functions in the process of the hard event should ideally be computed, as well as their fluctuations in the quantum sense. We know that this will not be for tomorrow and in a sense all arguments posed in a way or another are by definition unphysical, even the ones of the authors.
In this sense, this may be useful to avoid to be too categoric in the conclusions that this or that does not follow a good rationale, because this is also true for the conclusions of the paper.

Report

Based on my above arguments, I think the the criteria for a publication are met with no ambiguity.

Requested changes

no change

  • validity: high
  • significance: high
  • originality: good
  • clarity: top
  • formatting: excellent
  • grammar: excellent

Author:  Lucian Harland-Lang  on 2021-08-18  [id 1686]

(in reply to Report 1 on 2021-05-21)

We thank the author for their detailed and thoughtful comments, and take on board their general points made in the 'Weaknesses' section.

---

## Round 1 · Referee Report · Jean-Rene Cudell (Referee 2) · 2021-6-6

Report

This paper is topical, and I believe it is correct. It shows that the usual "solution" to the overshoot of muon data by theoretical predictions is not correct. After precisely quantifying the discrepancy, it suggests possible leads towards a solution. I think it should be published in its present form.
  • validity: top
  • significance: good
  • originality: good
  • clarity: high
  • formatting: excellent
  • grammar: excellent

Author:  Lucian Harland-Lang  on 2021-08-18  [id 1687]

(in reply to Report 2 by Jean-Rene Cudell on 2021-06-06)

Thank you Jean-Rene for taking the time to read and review our paper, and for your positive feedback.

---

## Round 1 · Referee Report · Anonymous (Referee 3) · 2021-7-3

Report

This paper describes the role of hadron-hadron interactions in inelastic photon-initiated processes ay the LHC.

In general, the paper is well written, detailed and describes new results. It definitely deserves publication once the few questions that I have are answered.

The authors discuss the impact of an impact parameter cut imposed on photon-induced processes in the Monte Calo when compared with the ATLAS measurement. In Fig. 1, I did not fully understand why the blue curve goes above the red one at high Q^2 when one imposes the impact parameter cut. In Table 1, I am wondering if the authors could comment about potential cuts at 2R_A, 3R_A, 4 R_A since the difference between 29.9 and 38.9 is quite large. Do Starlight and Superchic without survival probability also lead to the same results?

On page 9, it might be good to add a plot showing the dependence on the impact parameter cut or redo Fig.1 for different impact parameter cuts. I think this would illustrate better the discussion. Which parameter cut would then describe the ATLAS data best?

I am also wondering if the survival effect is indeed in “competition” with the impact parameter cut. It means that the survival probability will depend on the chosen impact parameter cut. In the same line, is it possible to use an impact parameter cut that is not a sharp cut, but some cuts dependent on kinematics (like the survival probability that depends on kinematics)?

There was also a recent observation by CMS that there is no further suppression of jet get jet events due to survival probabilities between 7 and 13 TeV, contrary to expectations. I am wondering of the situation is not even more complicated, and related to additional soft color exchanges that might modify the concept (and calculations) of survival probabilities. “Survival probabilities” can also be dependent on impact parameters.

On page 12, the authors might want to discuss briefly the validity of the EPA approximation especially at high mass?

On page 2, the authors might want to quote a recent paper published about ALPs in heavy ions: https://arxiv.org/abs/1903.04151
  • validity: -
  • significance: -
  • originality: -
  • clarity: -
  • formatting: -
  • grammar: -

Author:  Lucian Harland-Lang  on 2021-08-18  [id 1688]

(in reply to Report 3 on 2021-07-03)

We thank the referee for their constructive comments, which we address below, as well as in the updated manuscript. Please note that we have in addition added footnote 1 to page 15 to address an issue that has come to our attention since the original submission.

  • In Fig. 1, I did not fully understand why the blue curve goes above the red one at high Q^2 when one imposes the impact parameter cut.

We hope this point is discussed sufficiently in the paragraph above Section 3 in the text. The key point is just that imposing a cut in impact paramater space requires the cross section when integrated over the photon q_t to be smaller, but for a particular point in q_t space this does not have to be the case.

  • In Table 1, I am wondering if the authors could comment about potential cuts at 2R_A, 3R_A, 4 R_A since the difference between 29.9 and 38.9 is quite large.

The aim here is to simply compare with what Starlight does, which is to impose a cut of b_t > R_A. However, as we discuss in the paper it is incorrect to impose such a cut, and the same would apply for the cuts (b_t > 2R_A and so on). We would like to avoid any impression at all that one can treat this cut as a tuneable parameter (encoding some model dependence) that one can explore in order to match the data. It is not; as we say it should not be imposed at all.

For that reason, we would not like to present such a comparison. We hope the referee understands our view on this point.

  • Do Starlight and Superchic without survival probability also lead to the same results?

Up to very small effects they should do - if one imposes the unphysical b_t > R_A cut in the Superchic case. However, having investigated this we do not believe it is possible to provide a cross section prediction from Starlight with survival effects turned off, and hence cannot make such a comparison.

  • On page 9, it might be good to add a plot showing the dependence on the impact parameter cut or redo Fig.1 for different impact parameter cuts. I think this would illustrate better the discussion. Which parameter cut would then describe the ATLAS data best?

As discussed above we cannot present such a comparison: there can be no question of which parameter describes the ATLAS data better. Such a cut is unphysical and so should not be imposed at all. The only purpose of our discussion here is to indicate the impact that the b_t > R_A cut (given this what is imposed in e.g. Starlight) has.

  • I am also wondering if the survival effect is indeed in “competition” with the impact parameter cut. It means that the survival probability will depend on the chosen impact parameter cut.

It is certainly true that the relative impact of the survival factor will vary dependening on whether the unphysical b_t cut is imposed.

  • In the same line, is it possible to use an impact parameter cut that is not a sharp cut, but some cuts dependent on kinematics (like the survival probability that depends on kinematics)?

In practice yes one could implement such a smooth cut, by suitably modifying the theta function in (21). However there is no physical motivation to do this, as discussed above.

  • There was also a recent observation by CMS that there is no further suppression of jet get jet events due to survival probabilities between 7 and 13 TeV, contrary to expectations. I am wondering of the situation is not even more complicated, and related to additional soft color exchanges that might modify the concept (and calculations) of survival probabilities. “Survival probabilities” can also be dependent on impact parameters.

In our calculation we have accounted for the dependence of survival factor on the impact parameter. In reality, the expected difference in S^2 between 7 and 13 TeV is not too large, even for the jet production. For the photon induced reactions we consider in our paper, the dominant contribution comes from a larger impact parameters where the value of S^2 is close to 1. Hence, the corresponding energy dependence in our case will be rather weak.

  • On page 12, the authors might want to discuss briefly the validity of the EPA approximation especially at high mass?

We are not completely sure which aspect the referee has in mind here. The principle point as we understand it is that at higher masses the minimum photon Q^2 becomes larger, and this can have an important effect; however this is accounted for in our calculation.

  • On page 2, the authors might want to quote a recent paper published about ALPs in heavy ions: https://arxiv.org/abs/1903.04151

We have added this.

---

## Editorial Decision

resubmitted